# Can a Single Trial of a Thoracolumbar Myofascial Release Technique Reduce Pain and Disability in Chronic Low Back Pain? A Randomized Balanced Crossover Study

**DOI:** 10.3390/jcm10092006

**Published:** 2021-05-07

**Authors:** Luana Rocha Paulo, Ana Cristina Rodrigues Lacerda, Fábio Luiz Mendonça Martins, José Sebastião Cunha Fernandes, Leonardo Sette Vieira, Cristiano Queiroz Guimarães, Sílvia de Simoni Guedes Ballesteros, Marco Túlio Saldanha dos Anjos, Patrícia Aparecida Tavares, Sueli Ferreira da Fonseca, Murilo Xavier Oliveira, Mário Bernardo-Filho, Danúbia da Cunha de Sá-Caputo, Vanessa Amaral Mendonça, Redha Taiar

**Affiliations:** 1Postgraduate Program of Rehabilitation and Functional Performance (PPGReab), Federal University of Jequitinhonha and Mucuri Valleys (UFVJM), Diamantina 39803-371, Brazil; luana.r.p@hotmail.com (L.R.P.); lacerdaacr@gmail.com (A.C.R.L.); fmartins.ufvjm@gmail.com (F.L.M.M.); muriloxavier@gmail.com (M.X.O.); vaafisio@hotmail.com (V.A.M.); 2Academia Brasileira de Fascias, Juatuba 35675-000, Brazil; leosettefisio@gmail.com (L.S.V.); cristiano_fisioterapia@yahoo.com.br (C.Q.G.); guedesilvia@gmail.com (S.d.S.G.B.); 3Physiotherapy Departament, Una University Center, Divinópolis 35500-017, Brazil; tavaresaguiar@yahoo.com.br; 4Physiotherapy Departament, Federal University of Jequitinhonha and Mucuri Valleys, Diamantina 39803-371, Brazil; jscf1912@gmail.com (J.S.C.F.); suffonseca@hotmail.com (S.F.d.F.); 5Dinâmica Soluções em Saúde, Belo Horizonte 30210-590, Brazil; msaldanhadosanjos@yahoo.com.br; 6Laboratory of Mechanical Vibrations and Integrative Practices, State University of Rio de Janeiro, Rio de Janeiro 20550-013, Brazil; bernardofilhom@gmail.com (M.B.-F.); dradanubia@gmail.com (D.d.C.d.S.-C.); 7MATériaux et Ingénierie Mécanique (MATIM), Université de Reims Champagne-Ardenne, 51100 Reims, France

**Keywords:** fascia, chronic low back pain, myofascial release

## Abstract

Although manual therapy for pain relief has been used as an adjunct in treatments for chronic low back pain (CLBP), there is still the belief that a single session of myofascial release would be effective. This study was a crossover clinical trial aimed to investigate whether a single session of a specific myofascial release technique reduces pain and disability in subjects with CLBP. 41 participants over 18 years old were randomly enrolled into 3 situations in a balanced and crossover manner: experimental, placebo, and control. The subjects underwent a single session of myofascial release on thoracolumbar fascia and the results were compared with the control and placebo groups. The outcomes, pain and functionality, were evaluated using the numerical pain rating scale (NPRS), pressure pain threshold (PPT), and Oswestry Disability Index (ODI). There were no effects between-tests, within-tests, nor for interaction of all the outcomes, i.e., NPRS (η ^2^ = 0.32, F = 0.48, *p* = 0.61), PPT (η^2^ = 0.73, F = 2.80, *p* = 0.06), ODI (η^2^ = 0.02, F = 0.02, *p* = 0.97). A single trial of a thoracolumbar myofascial release technique was not enough to reduce pain intensity and disability in subjects with CLBP.

## 1. Introduction

Chronic non-specific low back pain (CLBP) is a condition characterized by pain, stiffness, and/or muscular tension [1] and is an important health problem throughout the world [2]. In CLBP, pain processing and modulation by the central nervous system may be altered. Manual therapy (MT) is a conservative intervention for treatment of CLBP [1,3,4]. Myofascial release is a form of MT, which involves the application of low-load and long-duration stretches to the myofascial complex. Although the mechanisms of action and effectiveness in individuals with CLBP are still unclear, myofascial release techniques are widely used by physical therapists in the management of CLBP, with the intent to restore the optimal length of the fasciae tissue, decrease pain intensity, and improve functionality [1,4,5,6,7,8,9]. Previous reports point to the effects of stabilization or global physical exercise in pain relief in subjects with CLBP with few studies for manual techniques alone or as an adjunct therapy [10,11,12,13,14].

Considering that previous studies recommended the adoption of interventions focused on the soft-tissues for the management of CLBP, there is still the belief among clinicians that an isolated session of MT, e.g., myofascial release, is effective in reducing pain intensity and disability [9,15]. As far as we know, only one study investigated the effects of an isolated myofascial release protocol on pain intensity and disability in patients with CLBP. Although the authors did not show clearly whether the improvement in pain intensity and disability was clinically relevant, they suggested the myofascial protocol that was used reduced these outcomes [4]. In this investigation, one of the areas chosen for intervention was the thoracolumbar fascia (TLF), which contains a great proportion of post-ganglionic sympathetic fibers and is densely innervated by free nerve endings, as low-threshold-mechanosensitive C fibers, which are responsive to MT [5,16] with repeated mechanical or biochemical stimulation.

The anatomical distribution of TLF layers, the aforementioned specificities of this tissue, and the support of the literature on the potential nociceptive function of TLF in the etiology of low back pain [17], allow for the inference that active movement of the trunk, such as flexion-extension, will increase the shear of the tissue and contribute to pain reduction and function [7,18,19,20,21,22,23,24,25]. Therefore, the present study aimed to verify if there is an immediate effect of a specific myofascial release technique on the TLF of individuals with CLBP, measured by pain intensity and disability, in comparison with that of control and placebo situations [23,25].

## 2. Materials and Methods

### 2.1. Study Design

This was a crossover open clinical trial performed between February and June 2019. The trial was registered with the Clinical Trials Government Identifier (ReBEC—reference number 8197). The current study conformed to the Consolidated Standards of Reporting Trials statement for reporting clinical trial studies. All subjects provided written informed consent to participate in this study, which was conducted in accordance with ethical principles for research involving humans (principles of the Declaration of Helsinki) and received approval from the Ethics and Research Committee of the Federal University of Jequitinhonha and Mucuri Valleys (reference number 3.435.537).

All the participants underwent three situations in a randomized and balanced order. The sequence of situations was randomized using a website (www.random.org, accessed on 8 July 2019). A familiarization with the experimental procedures was performed, followed by anamnesis (age, sex, and level of physical activity) and evaluation of the prognosis or risk profile using the STarT Back Screening Tool (SBST) questionnaire, which consists of nine items divided into physical and psychosocial subscales [26].

### 2.2. Study Populations

Subjects were recruited in the city of Diamantina, Minas Gerais, Brazil. Inclusion requirements and eligibility were men and women over 18 years old, with a medical diagnosis of CLBP or low back pain for more than 3 months, who obtained a minimum cut-off value of 2 points of pain by the Numerical Pain Rating Scale (NPRS) during data collection. Exclusion criteria were previous or scheduled surgeries in the torso or limbs; those with suspicion of severe fractures or pathologies (tumor, inflammation, infection, rheumatological disorder, aortic aneurysm); diagnosis of radiculopathy or neuropathy (with or without spinal canal stenosis with proof of magnetic resonance imaging—MRI); structural deformity in the spinal column; spondyloarthropathy, disabling pain and physical disability that would make it impossible to perform the study procedures; use of painkillers or anti-inflammatory medicines 48 h before the first test phase or during the study; neurological or psychiatric disorder; and presence or suspicion of pregnancy. In addition, the subjects that self-reported physical activity levels equal to or greater than those recommended by the American College of Sports Medicine (ACSM) were excluded [27,28,29].

### 2.3. Sample Size Calculation

The sample size was estimated by the GPower^®^ program (Franz Faul, Universität Kiel, Kiel, Germany), version 3.1.9.2. For this, we used a priori analysis, with ANOVA for comparisons between groups for the variable NPRS [30]. The effect size was calculated from the difference in the means with standard deviation within each group of 0.9. Thus, considering an effect size of 0.42, power of 0.80%, and alpha error 5%, the sample size was estimated at 41 volunteers. There were no withdrawals, so there was no need to analyze the data with the intention to treat.

A total of 52 subjects agreed to participate, of them, 10 individuals gave up or did not attend the test site on the scheduled date and 1 individual did not meet the described inclusion criteria. Therefore, 41 subjects participated in all stages of the research (Figure 1). Of the 41 individuals, 61% were women and 56% declared themselves sedentary. In addition, the median age was 36 years and the self-reported pain averaged 3.4 to 3.7 points. Among the subjects, 56% were classified in terms of poor prognosis in the SBST as low risk (Table 1). 

### 2.4. Outcome Measures

The measurements of pain were the primary outcome. We used the Pressure Pain Threshold (PPT) and NPRS as the instruments to measure pain. As a secondary outcome, the Oswestry Disability Index questionnaire (ODI—version 2.0, Department of Nursing, Faculty of Medical Sciences, State University of Campinas, Campinas, SP, Brazil) evaluated the prognosis and functionality of the subjects. All analyses were performed to compare the results before (pre-test) and immediately after (follow-up) each experimental situation.

For pain measurements, the NPRS was used to quantify the intensity of pain in all areas of the study, ranging from 0 to 10, with 0 classified as no pain and 10 as the worst pain imaginable. This instrument is proven to be a competing and valid predictor of pain intensity [31]. The pain threshold was assessed using a PPT device (FDX Series Force Gage, Wagner Instruments, Greenwich, CT, USA), with a graduation capacity of 50 × 0.05 lbf, 800 × 0.5 ozf, 25 × 0.02 kgf and 250 × 0.2 N, and a 1 cm^2^ plunger connected to a mechanical force gauge that indicates the pressure applied at the marked location. The plunger of the device was positioned perpendicularly to the paravertebral muscles, respecting the proximity of 2 cm laterally to the midline between the L3–L4 spinous process. The pressure was applied progressively and perpendicular to the skin, with an average of 1 kg/cm²/s, until the volunteer signaled the onset of pain or discomfort. After signaling, the device was removed from contact with the individual’s body and the Newton’s quantification was noted [32,33]. Initially, a familiarization was performed on the anterior muscles of the forearm. For the assessment, after the participants were positioned for the interventions, the point for assessed was marked with a pen and three measurements were taken at the same site (pre-test and follow-up), with a 30 s interval between them. The mean of the three measurements was used for analysis [1,34].

The ODI Version 2.0 questionnaire [35] was used to quantify the disability caused by low back pain in daily activities. Scores were given from 0% (no dysfunction; independence) to 100% (lowest level of functionality; total dependence), divided into 5 levels, where the first described no limitation and the others described limitation or inability to function. The total score is the percentage value calculated by the following equation: Total score = (Σ item score/50) × 100. With all 10 questions answered, the total score was divided by 50 (10 × 5). The published minimally clinically important difference (MCID) values for ODI is 12.88 (sensitivity 88%, specificity 85%) [36]. It must be taken into consideration that the ODI is a questionnaire with a one-dimensional factorial structure. Important psychometric properties and internal consistency, and their ability to measure functional limitations were considered to be very accurate according to the actual level of severity of the dysfunction experienced by the assessed subjects [37].

### 2.5. Intervention

Forty-one of the fifty-two volunteers were selected after the initial screening and exclusion through the eligibility criteria. All were enrolled in the three situations: (1) control (C), (2) experimental (Exp), and (3) placebo (Plac). In our work, a single trained and experienced therapist, a member of the Brazilian Academy of Fasciae (ABSFascias), applied the technique in all situations. Due to the design of our study, only the assessor was blinded. However, all the subjects performed the three situations on consecutive days respecting a minimum interval of 24 h in order to minimize the potential bias due to the high level of adaptability of the connective tissue [3].

#### 2.5.1. Control

The subjects were instructed to remain in the supine position for five minutes on a stretcher to minimize the effects of tension forces or stimuli on the tissue, without making movements, to mimic the time spent in the act of performing the intervention.

#### 2.5.2. Experimental

The subjects underwent a single session of the new approach to be tested to release the TLF in a sedestation position with feet supported and the thoracolumbar region properly undressed. The trunk flexion goniometry of each participant was performed and the value of 30° was marked with a barrier to limit the necessary movement during the technique. The trained researcher positioned their hands on all participants without sliding over the skin or forcing the tissue, with the cranial hand close to the last rib and at the T12–L1 level on the right side of the individual’s body and the caudal hand on the ipsilateral side between the iliac crest and the sacrum. Then, the researcher caused a slight traction in the tissues by moving their hands away from each other in a longitudinal direction. Then, the participant was instructed to perform five repetitions of active trunk flexion-extension (30°), while the researcher followed the movement with both hands simultaneously positioned, without losing the initial tissue traction and position (Figure 2). The same technique and the same number of repetitions of active trunk flexion-extension were repeated with the researcher’s hands positioned on the opposite sides. This technique lasted approximately five minutes [3,38,39].

#### 2.5.3. Placebo

The subjects were not submitted to the technique of manual thoracolumbar fascia release, but they slowly performed ten repetitions of active trunk flexion-extension (30°) in the same position as the experimental situation. Due to the fact that touch can provide not only well-recognized discriminative input to the brain, but also an affective input, there was no touch from the researcher at this stage [16].

### 2.6. Statistical Analysis

Data were described as mean (95% confidence interval). The normality and homogeneity of the variables were assessed by the Shapiro-Wilk test and the Levene test, respectively. The effects and interactions in the two moments (pre and post) and three situations (C, Plac, Exp) were evaluated by factorial variance analysis (ANOVA 2 × 3) allowing comparisons between-tests, within-tests, and with interaction. The Tukey test was used as a Post Hoc for multiple comparisons of means. The level of significance adopted for all tests was 5%. Estimates of effect size and power were calculated using the GPower^®^ program version 3.1 (Heinrich-Heine-University Düsseldorf, Germany).

## 3. Results

There was no statistical differences between-tests, within-tests, nor interaction. The minimal detectable change (MDC: post-pre) for PPT was 3% (−1.14 points) to the control, 8% (+2.45 points) to the placebo, and 13% (+4.28 points) to the experimental. The MDC for NPRS was 0.005% to the control, 0.09% to the placebo, and 0.04% to the experimental, i.e., a change less than 0.3 points in the NPRS. The MDC for ODI was 13% (+2.22 points) to the control, 2% (+0.19 points) to the placebo, and 1% (0.03 points) to the experimental (Table 2).

## 4. Discussion

The aim of our work was to verify the immediate effect of a single specific thoracolumbar myofascial release technique in individuals with CLBP, concerning pain intensity and disability, in comparison with control and placebo situations. The main findings show that our dosage and specific technique may not have been enough to provide effects on pain intensity and disability in individuals with CLBP. An important aspect that might explain our findings is that the myofascial techniques found in the literature are heterogeneous regarding duration, frequency, and intensity [3,4]. In this sense, a previous study applied four sessions of myofascial treatment, each lasting 40 min, compared to a sham therapy [4]. Ajimsha and colleagues (2014) [3] also used a myofascial technique of 40 min applied in several sites, while in our work the technique lasted five minutes and was applied in only one tissue in addition to a trunk movement.

Our findings demonstrated that although six participants showed a reduction in the NPRS consistent with the MCID [31,40,41], the magnitude of the subjects’ pain measured by NPRS was not statistically significant between-test, within-tests, nor interaction. In addition, regarding the PPT measure, no significant difference was observed between-tests, within-tests, nor with interaction according to the MCID of the PPT, which in the literature considered a change of 15% [42]. In addition, the average value found in the present study was around 34 N/cm^2^ (3.5 Kgf/cm^2^), representing a threshold lower than that reported in the literature [33,43]. According to Fischer (1987) [43], asymptomatic people are expected to report pain or discomfort to PPT test when reaching 55 N/cm^2^ (5.6 kgf/cm^2^) in men and 37 N/cm^2^ (3.8 kgf/cm^2^) in women, while Pöntinen (1998) [33] found a pain threshold of 39 N/cm^2^ (4.0 kgf/cm^2^) in a research with participants with CLBP.

These results probably have suffered interference related to the heterogeneity of the studied population regarding the poor prognosis by SBST, with 56% of the participants classified as low risk, 24.4% medium risk and 19.5% high risk. In addition, four participants of high risk and two of medium risk showed a significant improvement at NPRS in the intervention situation. According to Fritz et al. (2011) [26], there is a relationship between the risk categories and the magnitude of the participants’ improvement at the end of the analysis, in which individuals with a higher risk of poor prognosis, presented a greater report of pain attenuation [26].

In terms of the heterogeneity of our sample, some differences related to age and gender must be considered. The reduction in shear strain and increase in thickness of the posterior layer of the TLF in CLBP patients are more significant in the male gender, with a positive correlation between shear strain and low back pain duration [24]. In this sense, range of motion and physical function, body composition, fat distribution pattern, hormonal factors, or structural and/or movement pattern differences between males and females also might need to be considered [24,44]. With aging, biochemical, cellular, and functional changes occur in addition to changes in the structure of the extracellular matrix, and these age-related alterations in fascial tissues lead to densification and fibrosis (thixotropic behavior of hyaluronic acid and collagen synthesis increase thickness) contributing to pain occurrence [17,44].

Concerning functionality outcome, the ODI questionnaire revealed no significant difference between-tests, within-tests, nor with interaction. Lauridsen et al. (2006) [27] evaluated the response capacity of the ODI and MCID of ODI for patients with CLBP and pointed out that a MCID around 12–13 points is clinically important. In addition, in the present study, the interval between applications of the questionnaires was 24 h, which probably influenced the results obtained as it reduced confounding and memory factors, since the objective was to evaluate the effectiveness of the technique. The article on the development of the Brazilian version of the ODI [35] discusses this interference of the retest time in the results and also states that a longer interval improves the chances of reducing the final percentage due to the influence of the natural course of the associated CLBP symptoms.

The time factor calls into question the need to follow-up, because in addition to the natural course of low back pain, it is known that a short-term MT intervention can improve pain and disability but without retention effects after three-months follow-up [1,4]. Nevertheless, habitual loading will result in a high rate of collagen synthesis in a basal state simply as a result of a constant effect of loading from the previous 24–48 h. Magnusson et al. (2010) [45] observed that after cessation of exercise and up to 18–36 h thereafter, there is a negative net balance in collagen levels, whereas the balance is positive for up to 72 h after exercise. However, the connective tissue requires a certain restitution period, since, without sufficient rest, a continuous loss of collagen is likely to occur, which might render the tissue vulnerable. Habitual physical exercises thus results in a higher turnover of collagen, whereas inactivity lowers collagen synthesis and also the turnover [45].

Considering these facts, it is accepted that CLBP has biological, psychological, and social components in several different extensions in addition to the biomechanics involved in its development. It must be considered that each subject has different painful experiences and different outcomes throughout life, where multiple areas of the brain are activated during a pain experience [18,46]. Moreover, these central areas have other primary functions, i.e., movement execution, sensory location, and emotional awareness, which are overloaded in chronic pain, and may explain the emergence of psychosocial problems among other motor and sensory changes that are beyond the scope of this study [46,47,48].

The multicausality and variety of outcomes presented by chronic pain, in addition to the non-linear interaction arising from the complexity of the interaction of causal factors, may explain the fact that a single session of myofascial mobilization was not enough to modify the threshold and intensity of the pain and functional capacity [49,50,51]. Additionally, a rehabilitation program focused not only on tissue but also on complementary issues such as exercise, pain education, and behavioral strategies is important [47,49,52,53,54].

### Limitations

The results probably have been influenced by different factors such as the heterogeneity of the studied population regarding the poor prognosis by SBST and the interval between experimental conditions and measurement time points. In addition, other factors, such as the myofascial technique applied compared with other thoracolumbar myofascial approaches, the characteristics of the sample regarding age and gender, or the pain intensity of the sample at baseline (3.7/10), could also have influenced the results. Finally, trunk flexion-extension range of motion and psychological aspects were not evaluated in our work and should be considered.

Another limitation is that there is still no way to quantify the pressure applied by the therapist’s hand during manual therapy techniques, making it difficult to reproduce the studied technique. Much of the effects of MT relies on the ability of the physical therapist to sense the changes in the tissue. The biological effects of touch can also change the effectiveness of the treatment, depending on the state of either the physical therapist or the patient. This variability means that interrater reliability is low, and therefore prevents MT from being considered evidence-based [9].

## 5. Conclusions

This study provides evidence that a single trial of thoracolumbar myofascial release technique was not enough to reduce pain and disability in subjects with CLBP. Further investigations combining other interventions with myofascial mobilization and those with prolonged treatments are required. The mechanisms underlying these responses merit further investigation. We recommend the design of studies that take into account the bio-psychosocial aspects of individuals with CLBP in addition to analyzing the effects of myofascial mobilization at the structural level of the tissue.

## Figures and Tables

**Figure 1 jcm-10-02006-f001:**
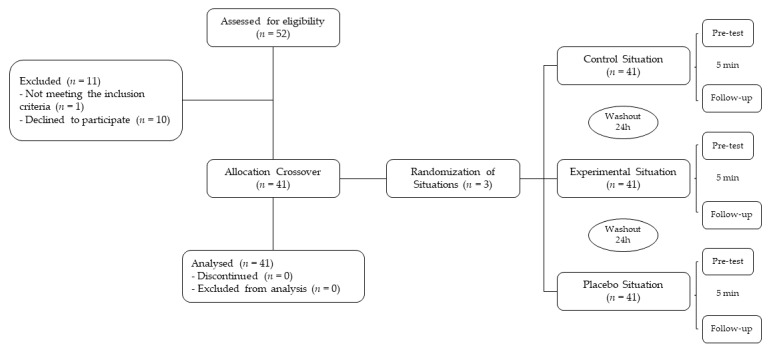
Flow of participants throughout in all stages of the research.

**Figure 2 jcm-10-02006-f002:**
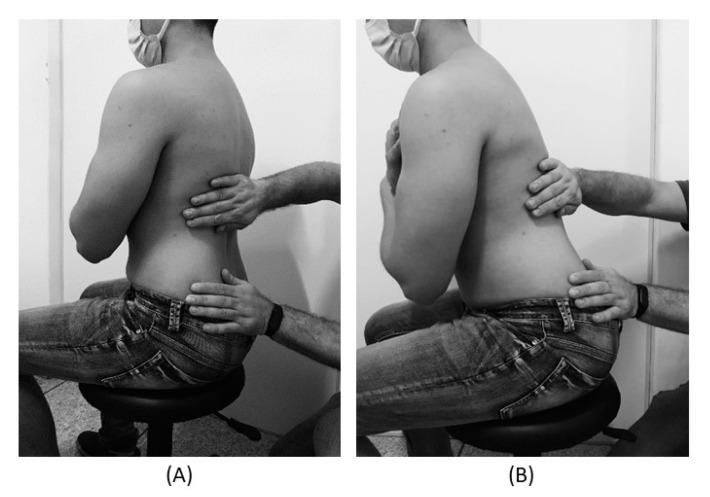
(**A**) Trunk position (90°) and researcher’s hands contact at the beginning of the technique. (**B**) 30° trunk flexion and researcher’s hands contact at the end of the technique.

**Table 1 jcm-10-02006-t001:** Characteristics of the participants at baseline.

Characteristics	Subjects (*n* = 41) Mean (95% CI *) or %
Age (years)	36 (22–50)
Gender, women (%)	60.98
Sedentary (%)	56.09
NRPS ^1^ baseline (0–10)	3.68 (1.39–5.97)
PPT ^2^ (N/cm^2^)	34 (16.57–51.43)
FTF ^3^ (cm)	13 (2–24)
SBST ^4^—prognosis (%) Low risk Medium risk High risk	56.1 24.4 19.5

* CI: confidence interval; ^1^ NRPS: visual numeric pain scale; ^2^ PPT: pain pressure threshold; ^3^ FTF: fingertip-to-floor; ^4^ SBST: STarT Back Screening Tool.

**Table 2 jcm-10-02006-t002:** Statistical results between-tests, within-tests and interaction for outcomes.

Outcomes	Control Mean (95% CI)	Placebo Mean (95% CI)	Experimental Mean (95% CI)	Between-Tests				Within-Tests				Interaction			
				*p*	F	η^2^	Power	*p*	F	η^2^	Power	*p*	F	η^2^	Power
PPT (N/cm^2^) PT	37.25 (32.63–41.86)	29.37 (23.93–34.81)	30.38 (24.70–36.06)	0.40	0.90	0.47	0.99	0.56	0.34	0.25	0.90	0.06	2.80	0.73	1.00
PPT (N/cm^2^) FU	36.11 (30.90–41.31)	31.82 (26.12–37.52)	34.66 (28.01–41.32)												
NPRS (score) PT	3.41 (2.69–4.12)	3.80 (3.06–4.53)	3.00 (2.30–3.69)	0.06	2.79	0.73	1.00	0.80	0.06	0.05	0.25	0.61	0.48	0.32	0.97
NPRS (score) FU	3.43 (2.76–4.09)	3.48 (2.73–4.22)	3.14 (2.38–3.89)												
ODI (%) PT	15.82 (12.91–18.74)	17.51(14.42–20.59)	19.26 (16.29–22.23)	0.007	5.01	0.83	1.00	0.73	0.11	0.07	0.31	0.97	0.02	0.02	0.11
ODI (%) FU	18.04 (15.00–21.09)	17.70 (14.65–20.76)	19.29 (16.22–22.36)												

CI: confidence interval; PPT: pressure pain threshold (N: Newtons); NPRS: visual numeric pain scale; ODI: Oswestry Disability Index; PT: pre-test; FU: follow-up. η^2^: Eta partial. Sample size = 41. Factorial ANOVA (2 × 3).

## Data Availability

The data presented in this study are available on request from the corresponding author. The data are not publicly available due to the privacy guarantee of the data collected individually.

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
