# Peer review of "Can a Single Trial of a Thoracolumbar Myofascial Release Technique Reduce Pain and Disability in Chronic Low Back Pain? A Randomized Balanced Crossover Study"

_jcm, 2021, doi:10.3390/jcm10092006_

Round 1
Reviewer 1 Report
MANUSCRIPT NUMBER: jcm-1178797
Thank you very much for considering me as a reviewer for this article. The present study analyses the immediate effects of a single thoracolumbar MFR technique on pain and disability in chronic LBP patients, compared to control and placebo interventions. The trial has been registered with the Clinical Trials Identifier (ReBEC – reference number 8197) and has been approved by the concerned Ethics Committee. However, there are some unclear aspects. Please check the following review feedback and revise if it is needed.
INTRODUCTION
- Lines 48-49: why references 2 and 14 are cited here to support the belief about effects of an isolated session of manual therapy? I do not think that the mentioned authors develop this idea in the cited studies.
- The technique developed in this study is an unusual thoracolumbar myofascial release technique as it combines the mechanical stimulation of the fascia induced by the hands of the investigator with the active trunk movement. In lines 63-64, the authors suggest that this movement of the trunk together with the myofascial release applied concomitantly may contribute to the normalization of the flexion-relaxation response. However, no variable related with flexion-relaxation phenomenon has been registered in this study and, in addition, no further comments about flexion-relaxation response are included in the discussion section. Consequently, which is the sense about introducing this idea?
- Line 68: as the intervention applied in the current study was a single thoracolumbar myofascial release technique it would be more accurate to use “specific myofascial release technique on TLF” instead of “specific myofascial release protocol on TLF”. To avoid confounding, please change this expression along the manuscript.
METHODS
- In the inclusion criteria it is stated that volunteers had to score at least 2-3 points of pain by the NPRS. I think that the authors should determine a specific cut-off value for this pain measurement. Which was the minimum value needed?
- Line 92-93: It is also an inclusion criteria to have a low or sedentary level of physical activity. How was this condition assessed? Somewhere in the text it is said that participants self-reported their level of physical condition. However, when was this level considered as low, medium or high? Please, explain it. Moreover, if a low or sedentary level of physical activity was required as an inclusion criteria, it seems a rebuttal that the 44% of the sample considered themselves active as it is stated in lines 255-256.
- Line 98: How was assessed the movement limitation in order to discard the participants? Was any exploration performed?, Was any instrument used?, was it also a self-reported item? Which was the criterion to consider a significant movement limitation?
- Regarding the method for the pain threshold assessment, was only one measure registered pre and post intervention, respectively? There were any repetitions performed for each measurement time point?
- Line 175: In the description of the intervention applied in the experimental group it is said that “the participant was instructed to perform five repetitions of active trunk flexion-extension (30º)”. How was the range of the trunk movement controlled during the intervention? Was this myofascial release technique previously used in other studies? I recommend, for a better understanding, to add an image with the initial position of the trunk showing the hands contact and other image with the final position of the trunk at the end of the technique.
- The placebo intervention just consisted in performed the repetitions of active trunk flexion-extension, however no manual contact was provided to the participants, simultaneously. Given the characteristics of the technique applied to the experimental group, is there any reason why the authors did not consider providing a manual contact, without exerting tissue traction, in the placebo group?. This form of placebo for myofascial techniques is widely used among studies (Tozzi et al. (2011), Ajimsha et al. (2014), Arguisuelas et al. (2017)).
- Who did perform the interventions? Which experience did the investigator have on myofascial release?
- No information is provided about the blinding. If done, please describe how the blinding was accomplished. If not, the authors should value the risk of bias regarding the absence of blinding.
DISCUSSION
- My main concern in the discussion is about the generalisability of the trial findings. I strongly agree with the authors when they say that the results probably have been influenced by different factors such as the heterogeneity of the studied population regarding the poor prognosis by SBST, the interval between experimental conditions and disability measurement time points or the level of the physical activity self-reported by the sample. In addition, other factors no mentioned by the authors, such as the myofascial technique applied vs other thoracolumbar myofascial approaches, the characteristics of sample regarding age and gender or the pain intensity of the sample at baseline (3.7/10), could also influence the results. According to this, it is necessary to clarify that the findings of the study arise from specific population and methodologic conditions. Moreover, the affirmation done in lines 207-209 should be expressed in a more conservative manner.
Author Response
Journal of Clinical Medicine
Ana Gonzalez, PhD
Editorial Board Member
Dear Editor,
We are pleased that the reviewers are favorable to the publication of our manuscript entitled “Can a single trial of a thoracolumbar myofascial release technique reduce pain and disability in chronic low back pain? Randomized balanced cross-over study”. We send enclosed point-by-point comments to reviewers about all remaining concerns and requests for editing (modifications highlighted).
Yours sincerely,
Prof R. Taiar
https://orcid.org/0000-0002-0227-3884
https://pubmed.ncbi.nlm.nih.gov/?term=taiar+r
https://www.scopus.com/authid/detail.uri?authorId=15823162100
https://www.researchgate.net/profile/Redha_Taiar
https://scholar.google.fr/citations?user=2FP-NPQAAAAJ&hl=fr&oi=ao
Ana Cristina Rodrigues Lacerda, PhD
Universidade Federal dos Vales do Jequitinhonha e Mucuri (UFVJM)
Mail address: Campus JK - Rodovia MGT 367 – Km 583, nº 5000. Bairro Alto da
Jacuba, CEP 39100-000. Diamantina, Brazil
Phone number: +55 (38) 3532-6981
Email: lacerdaacr@gmail.com / lacerda.acr@ufvjm.edu.br
Reviewer Comments:
#Reviewer 1
Lines 48-49: why references 2 and 14 are cited here to support the belief about effects of an isolated session of manual therapy? I do not think that the mentioned authors develop this idea in the cited studies.
We verified the references and changed accordingly.
The technique developed in this study is an unusual thoracolumbar myofascial release technique as it combines the mechanical stimulation of the fascia induced by the hands of the investigator with the active trunk movement. In lines 63-64, the authors suggest that this movement of the trunk together with the myofascial release applied concomitantly may contribute to the normalization of the flexion-relaxation response. However, no variable related with flexion-relaxation phenomenon has been registered in this study and, in addition, no further comments about flexion-relaxation response are included in the discussion section. Consequently, which is the sense about introducing this idea?
We agree with the reviewer that the additional information about flexion-relaxation response it is meaningless in our study since the main outcome was pain and not trunk flexion-extension range of motion. Thus, we removed the information related with flexion-relaxation phenomenon in the introduction and in the discussion.
In addition, we inserted, as perspective, future studies should consider the evaluation of other variables, e.g., trink flexion-extension range of motion.
Line 68: as the intervention applied in the current study was a single thoracolumbar myofascial release technique it would be more accurate to use “specific myofascial release technique on TLF” instead of “specific myofascial release protocol on TLF”. To avoid confounding, please change this expression along the manuscript.
We thank and agree with the reviewer. Therefore, we made the changes as suggested.
In the inclusion criteria it is stated that volunteers had to score at least 2-3 points of pain by the NPRS. I think that the authors should determine a specific cut-off value for this pain measurement. Which was the minimum value needed?
We agree with the reviewer. To clarify this information in the work, we changed to 2 points for pain in the NPRS.
Line 92-93: It is also an inclusion criterion to have a low or sedentary level of physical activity. How was this condition assessed? Somewhere in the text it is said that participants self-reported their level of physical condition. However, when was this level considered as low, medium or high? Please, explain it. Moreover, if a low or sedentary level of physical activity was required as an inclusion criterion, it seems a rebuttal that the 44% of the sample considered themselves active as it is stated in lines 255-256.
In fact, we have adopted as an exclusion criterion with self-reported physical activity lev-el below than recommended by American College of Sports Medicine. According to the ACSM, adults should do at least 150–300 minutes of moderate-intensity aerobic physical activity, or at least 75–150 minutes of vigorous-intensity aerobic physical activity, or an equivalent combination of moderate- and vigorous-intensity activity throughout the week, for substantial health benefits.
We also removed the text below from the discussion: “In this work, the performance of physical exercises, according to the level of physical activity self-reported by the sample, may also have interfered in the outcomes. 44% of the sample considered themselves active, but doing vigorous exercise less than 3 times per week, or moderate physical activity less than 5 times per week, this information do not meet the guidelines of the American College of Sports Medicine for health benefits [26]. Even though the level of physical activity is not significant, this information may have influenced the results.”
Line 98: How was assessed the movement limitation in order to discard the participants? Was any exploration performed? Was any instrument used? Was it also a self-reported item? Which was the criterion to consider a significant movement limitation?
“In addition, the subjects obtained an average of 13 cm of trunk flexion (FTF) compatible with previous data for patients with low back pain [38,39] (Table 1).”
Regarding the method for the pain threshold assessment, was only one measure registered pre and post intervention, respectively? There were any repetitions performed for each measurement time point?
The study design was crossover and all the participants underwent three situations in a randomized and balanced order. All analyzes were performed to compare the results before (pre-test) and immediately after (follow-up) of each situation. We modified figure 1 to facilitate the understanding of the design and flow of participants through the trial.
Line 175: In the description of the intervention applied in the experimental group it is said that “the participant was instructed to perform five repetitions of active trunk flexion-extension (30º)”. How was the range of the trunk movement controlled during the intervention? Was this myofascial release technique previously used in other studies? I recommend, for a better understanding, to add an image with the initial position of the trunk showing the hands contact and other image with the final position of the trunk at the end of the technique.
We agree with the reviewer and inserted into the article an explanation describing the range of the trunk movement and an image (figure 2) as suggested.
Figure 2. (A) Trunk position and researcher’s hands contact at the beginning of the technique. (B) Trunk position and researcher’s hands contact at the end of the technique.
The placebo intervention just consisted in performed the repetitions of active trunk flexion-extension, however no manual contact was provided to the participants, simultaneously. Given the characteristics of the technique applied to the experimental group, is there any reason why the authors did not consider providing a manual contact, without exerting tissue traction, in the placebo group?. This form of placebo for myofascial techniques is widely used among studies (Tozzi et al. (2011), Ajimsha et al. (2014), Arguisuelas et al. (2017)).
To explain the non-use of touch at PLAC, we inserted the sentence as follow: “Due the the fact that the touch can provide not only well-recognized discriminative input to the brain, but also an affective input, there was no touch from the researcher at this stage.”
Who did perform the interventions? Which experience did the investigator have on myofascial release?
In our work, a single trained and experienced therapist, member of the Brazilian Academy of Fasciae (ABFascias), was responsible for applying the technique in all situations.
No information is provided about the blinding. If done, please describe how the blinding was accomplished. If not, the authors should value the risk of bias regarding the absence of blinding.
In our work, only the assessor was blinded. Thus, the assessor did not know which sequence of randomization the subjects had been allocated to. “Due to the design of our study, the subjects and therapist were not “blind”. However, all the subjects performed the three situations on consecutive days respecting a minimum interval of 24 hours in order to minimize the potential bias due to the high level of adaptability of the connective tissue [3].”
My main concern in the discussion is about the generalisability of the trial findings. I strongly agree with the authors when they say that the results probably have been influenced by different factors such as the heterogeneity of the studied population regarding the poor prognosis by SBST, the interval between experimental conditions and disability measurement time points or the level of the physical activity self-reported by the sample. In addition, other factors no mentioned by the authors, such as the myofascial technique applied vs other thoracolumbar myofascial approaches, the characteristics of sample regarding age and gender or the pain intensity of the sample at baseline (3.7/10), could also influence the results. According to this, it is necessary to clarify that the findings of the study arise from specific population and methodologic conditions. Moreover, the affirmation done in lines 207-209 should be expressed in a more conservative manner.
We agree with the reviewer inserted all the cited points in the discussion. We inserted in “item 4.1 Limitations” that the psychological aspects were not taken into account in our work and should be considered as perspective since catastrophism have a very strong component in low back pain.

Reviewer 2 Report
The paper presents a correct methodology and is is well organized. However, I advise a grammar check throughout all the paper.
The discussion, results and conclusions are well presented.
I advise authors to improve the introduction in order to explain more extensively the effect of myofascial techniques in nonspecific thoracolumbar pain.
A limitation of the study : Psychological aspects were not taken into account. Has a previous psychological assessment scale been carried out? Aspects such as expectations or catastrophism have a very strong component in thoracolumbar pain. Therefore, the expectations of the patient and their psychological situation should have been considered when establishing the inclusion and exclusion criteria.
Author Response
Journal of Clinical Medicine
Ana Gonzalez, PhD
Editorial Board Member
Dear Editor,
We are pleased that the reviewers are favorable to the publication of our manuscript entitled “Can a single trial of a thoracolumbar myofascial release technique reduce pain and disability in chronic low back pain? Randomized balanced cross-over study”. We send enclosed point-by-point comments to reviewers about all remaining concerns and requests for editing (modifications highlighted).
Yours sincerely,
Prof R. Taiar
https://orcid.org/0000-0002-0227-3884
https://pubmed.ncbi.nlm.nih.gov/?term=taiar+r
https://www.scopus.com/authid/detail.uri?authorId=15823162100
https://www.researchgate.net/profile/Redha_Taiar
https://scholar.google.fr/citations?user=2FP-NPQAAAAJ&hl=fr&oi=ao
Ana Cristina Rodrigues Lacerda, PhD
Universidade Federal dos Vales do Jequitinhonha e Mucuri (UFVJM)
Mail address: Campus JK - Rodovia MGT 367 – Km 583, nº 5000. Bairro Alto da
Jacuba, CEP 39100-000. Diamantina, Brazil
Phone number: +55 (38) 3532-6981
Email: lacerdaacr@gmail.com / lacerda.acr@ufvjm.edu.br
#Reviewer 2
The paper presents a correct methodology and is well organized. However, I advise a grammar check throughout all the paper.
Our work was initially reviewed by a native language teacher. However, as we adjusted to the text, we sent it back to the teacher for further review.
The discussion, results and conclusions are well presented.
We appreciate the reviewer’s comment.
I advise authors to improve the introduction in order to explain more extensively the effect of myofascial techniques in nonspecific thoracolumbar pain.
We improved the explanation about the effect of myofascial techniques in nonspecific thoracolumbar pain (marked in yellow)
A limitation of the study: Psychological aspects were not taken into account. Has a previous psychological assessment scale been carried out? Aspects such as expectations or catastrophism have a very strong component in thoracolumbar pain. Therefore, the expectations of the patient and their psychological situation should have been considered when establishing the inclusion and exclusion criteria.
We inserted in “item 4.1 Limitations” that the psychological aspects were not taken into account in our work and should be considered as perspective since catastrophismo have a very strong component in thoracolumbar pain.

Round 2
Reviewer 1 Report
I would like to thank the authors the effort of improve the manuscript. Some questions have been clarified in the new version but not all of them. That is the reason I would like to go back to the following pending issues:
INTRODUCTION
- Lines 48-49: why references 2 and 14 are cited here to support the belief about effects of an isolated session of manual therapy? I do not think that the mentioned authors develop this idea in the cited studies.
Answer: We verified the references and changed accordingly.
Rw2: Effectively, the authors have changed the references numbers and added new authors. Again, I do not think that reference 4 is properly used here to support the belief about effects of an isolated session since this RCT developed a protocol of myofascial release techniques. Consequently, in the following lines when referring to this study (4), the correct expression would be “myofascial release protocol”, as it was written in the preceding version of the mansucript, rather than “myofascial release technique” as it currently stands.
METHODS
- Line 92-93: It is also an inclusion criteria to have a low or sedentary level of physical activity. How was this condition assessed? Somewhere in the text it is said that participants self-reported their level of physical condition. However, when was this level considered as low, medium or high? Please, explain it. Moreover, if a low or sedentary level of physical activity was required as an inclusion criteria, it seems a rebuttal that the 44% of the sample considered themselves active as it is stated in lines 255-256.
Answer: In fact, we have adopted as an exclusion criterion with self-reported physical activity lev-el below than recommended by American College of Sports Medicine. According to the ACSM, adults should do at least 150–300 minutes of moderate-intensity aerobic physical activity, or at least 75–150 minutes of vigorous-intensity aerobic physical activity, or an equivalent combination of moderate- and vigorous-intensity activity throughout the week, for substantial health benefits.
We also removed the text below from the discussion: “In this work, the performance of physical exercises, according to the level of physical activity self-reported by the sample, may also have interfered in the outcomes. 44% of the sample considered themselves active, but doing vigorous exercise less than 3 times per week, or moderate physical activity less than 5 times per week, this information do not meet the guidelines of the American College of Sports Medicine for health benefits [26]. Even though the level of physical activity is not significant, this information may have influenced the results.”
Rw2: Now it is not clear if the level of physical activity was an inclusion criteria, as it is stated in the manuscript, or an exclusion criteria, as it is said by the authors in the response letter. Please, explain it and change the information if needed.
- Line 98: How was assessed the movement limitation in order to discard the participants? Was any exploration performed?, Was any instrument used?, was it also a self-reported item? Which was the criterion to consider a significant movement limitation?
Answer: Although we measured trunk flexion using the findertip-to-floor (FTF) instrument to characterize the sample, we did not adopt the trunk flexion measure as an eligibility criterion. We inserted the results of the trunk flexion measurement in table 1, which refers to the characteristics of the participants and baseline. Moreover, we inserted the sentence as follows:
“In addition, the subjects obtained an average of 13 cm of trunk flexion (FTF) compatible with previous data for patients with low back pain [38,39] (Table 1).”
Rw2: If movement limitation was not, finally, considered as an exclusion criteria, I recommend to remove this information from this paragraph, as done in the present version. It is not necessary to insert lines 107-109 as they refer to the baseline characteristics of the sample and this data is shown in table 1.
- Regarding the method for the pain threshold assessment, was only one measure registered pre and post intervention, respectively? There were any repetitions performed for each measurement time point?
Answer: The study design was crossover and all the participants underwent three situations in a randomized and balanced order. All analyzes were performed to compare the results before (pre-test) and immediately after (follow-up) of each situation. We modified figure 1 to facilitate the understanding of the design and flow of participants through the trial.
Rw2: The design of the study was clear with previous figure 1, it is obvious that there are two measurement time point (pre and post intervention). Maybe, the question was not properly addressed. I wanted to ask if only one trial of pressure pain threshold was measured or different trials were perfomed and the mean of the trials in each measurement time point was then calculated, as done in other studies.
DISCUSSION
- My main concern in the discussion is about the generalisability of the trial findings. I strongly agree with the authors when they say that the results probably have been influenced by different factors such as the heterogeneity of the studied population regarding the poor prognosis by SBST, the interval between experimental conditions and disability measurement time points or the level of the physical activity self-reported by the sample. In addition, other factors no mentioned by the authors, such as the myofascial technique applied vs other thoracolumbar myofascial approaches, the characteristics of sample regarding age and gender or the pain intensity of the sample at baseline (3.7/10), could also influence the results. According to this, it is necessary to clarify that the findings of the study arise from specific population and methodologic conditions. Moreover, the affirmation done in lines 207-209 should be expressed in a more conservative manner.
Answer: We agree with the reviewer inserted all the cited points in the discussion. We inserted in “item 4.1 Limitations” that the psychological aspects were not taken into account in our work and should be considered as perspective since catastrophism have a very strong component in low back pain.
Rw2: I, honestly, expected that the authors develop a rationale in the discussion including all the possible factors influencing the results rather than copy and paste the above lines in the limitation section. I strongly think it would be very interesting and enriching.
On the other hand, I insist that the authors should be cautious with the expression “The main findings suggest demystifying the belief that an isolated MT trial, as well as the thoracolumbar myofascial release technique, reduces pain and improves functionality in subjects with CLBP”. There is a list of potential factors which would have influenced the results, even the authors recognize “It is possible to assume that our dosage and technique specific may not have been enough to provide effects on pain and disability.” (lines 261-263). The discussion should clarify that these are the results of the present study (with a specific methodologica

Author Response
Journal of Clinical Medicine
Ana Gonzalez, PhD
Editorial Board Member
Dear Editor,
We send enclosed point-by-point comments to reviewer - minor points (modifications highlighted).
Title: “Can a single trial of a thoracolumbar myofascial release technique reduce pain and disability in chronic low back pain? Randomized balanced cross-over study”.
Yours sincerely,
Redha Taiar, PhD
University of Reims, France
Reviewer Comments:
#Reviewer#
I would like to thank the authors the effort of improve the manuscript. Some questions have been clarified in the new version but not all of them. That is the reason I would like to go back to the following pending issues:
INTRODUCTION
- Lines 48-49: why references 2 and 14 are cited here to support the belief about effects of an isolated session of manual therapy? I do not think that the mentioned authors develop this idea in the cited studies.
Answer: We verified the references and changed accordingly.
Rw2: Effectively, the authors have changed the references numbers and added new authors. Again, I do not think that reference 4 is properly used here to support the belief about effects of an isolated session since this RCT developed a protocol of myofascial release techniques. Consequently, in the following lines when referring to this study (4), the correct expression would be “myofascial release protocol”, as it was written in the preceding version of the mansucript, rather than “myofascial release technique” as it currently stands.
Answer2: We agree with the reviewer. As suggested, we deleted the reference at this part and modified the expression “myofascial release technique” to “myofascial release protocol”.
METHODS
- Line 92-93: It is also an inclusion criteria to have a low or sedentary level of physical activity. How was this condition assessed? Somewhere in the text it is said that participants self-reported their level of physical condition. However, when was this level considered as low, medium or high? Please, explain it. Moreover, if a low or sedentary level of physical activity was required as an inclusion criteria, it seems a rebuttal that the 44% of the sample considered themselves active as it is stated in lines 255-256.
Answer: In fact, we have adopted as an exclusion criterion with self-reported physical activity level below than recommended by American College of Sports Medicine. According to the ACSM, adults should do at least 150–300 minutes of moderate-intensity aerobic physical activity, or at least 75–150 minutes of vigorous intensity aerobic physical activity, or an equivalent combination of moderate- and vigorous-intensity activity throughout the week, for substantial health benefits.
We also removed the text below from the discussion: “In this work, the performance of physical exercises, according to the level of physical activity self-reported by the sample, may also have interfered in the outcomes. 44% of the sample considered themselves active, but doing vigorous exercise less than 3 times per week, or moderate physical activity less than 5 times per week, this information do not meet the guidelines of the American College of Sports Medicine for health benefits [26]. Even though the level of physical activity is not significant, this information may have influenced the results.”
Rw2: Now it is not clear if the level of physical activity was an inclusion criteria, as it is stated in the manuscript, or an exclusion criteria, as it is said by the authors in the response letter. Please, explain it and change the information if needed.
Answer2: We apologize to the reviewer for the misinterpretation. The self-reported physical activity level was adopted as exclusion criteria. Thus, we excluded the subjects that self-reported physical activity level equal to or greater than that recommended by American College of Sports Medicine (ACSM) [27–29].
To clarify this information, we changed as follows:
“In addition, subjects that self-reported physical activity level equal to or greater than the recommended by American College of Sports Medicine (ACSM) were excluded [27–29].”
- Line 98: How was assessed the movement limitation in order to discard the participants? Was any exploration performed? Was any instrument used? Was it also a self-reported item? Which was the criterion to consider a significant movement limitation?
Answer: Although we measured trunk flexion using the findertip-to-floor (FTF) instrument to characterize the sample, we did not adopt the trunk flexion measure as an eligibility criterion. We inserted the results of the trunk flexion measurement in table 1, which refers to the characteristics of the participants and baseline. Moreover, we inserted the sentence as follows:
“In addition, the subjects obtained an average of 13 cm of trunk flexion (FTF) compatible with previous data for patients with low back pain [38,39] (Table 1).”
Rw2: If movement limitation was not, finally, considered as an exclusion criteria, I recommend to remove this information from this paragraph, as done in the present version. It is not necessary to insert lines 107-109 as they refer to the baseline characteristics of the sample and this data is shown in table 1.
Answer2: We thank the reviewer! We removed lines 107-109.
- Regarding the method for the pain threshold assessment, was only one measure registered pre and post intervention, respectively? There were any repetitions performed for each measurement time point?
Answer: The study design was crossover and all the participants underwent three situations in a randomized and balanced order. All analyzes were performed to compare the results before (pretest) and immediately after (follow-up) of each situation. We modified figure 1 to facilitate the understanding of the design and flow of participants through the trial.
Rw2: The design of the study was clear with previous figure 1, it is obvious that there are two measurement time point (pre and post intervention). Maybe, the question was not properly addressed. I wanted to ask if only one trial of pressure pain threshold was measured or different trials were perfomed and the mean of the trials in each measurement time point was then calculated, as done in other studies.
Answer2: We thank the reviewer! We inserted additional informations (highlighted in yellow) to clarify the addressed question.
Line 142-149: The pressure was applied progressively and perpendicular to the skin, with an average of 100g.sec-1, until the volunteer signaled the onset of pain or discomfort. After signaling, the device was removed from contact with the individual's body and Newton's quantification was noted [32,33]. Initially, a familiarization was performed on anterior muscles of the forearm. For the assessment, after positioned for the interventions, the point assessed was marked with a pen and three measuments were taken at site (pre-test and follow-up), with a 30-seconds interval between them. The mean of the three measuments was used for analysis [34,35].
DISCUSSION
- My main concern in the discussion is about the generalisability of the trial findings. I strongly agree with the
authors when they say that the results probably have been influenced by different factors such as the heterogeneity of the studied population regarding the poor prognosis by SBST, the interval between experimental conditions and disability measurement time points or the level of the physical activity self-reported by the sample. In addition, other factors no mentioned by the authors, such as the myofascial technique applied vs other thoracolumbar myofascial approaches, the characteristics of sample regarding age and gender or the pain intensity of the sample at baseline (3.7/10), could also influence the results. According to this, it is necessary to clarify that the findings of the study arise from specific population and methodologic conditions. Moreover, the affirmation done in lines 207-209 should be expressed in a more conservative manner.
Answer: We agree with the reviewer inserted all the cited points in the discussion. We inserted in “item 4.1 Limitations” that the psychological aspects were not taken into account in our work and should be considered as perspective since catastrophism have a very strong component in low back pain.
Rw2: I, honestly, expected that the authors develop a rationale in the discussion including all the possible factors influencing the results rather than copy and paste the above lines in the limitation section. I strongly think it would be very interesting and enriching.
On the other hand, I insist that the authors should be cautious with the expression “The main findings suggest demystifying the belief that an isolated MT trial, as well as the thoracolumbar myofascial release technique, reduces pain and improves functionality in subjects with CLBP”. There is a list of potential factors which would have influenced the results, even the authors recognize “It is possible to assume that our dosage and technique specific may not have been enough to provide effects on pain and disability.” (lines 261-263). The discussion should clarify that these are the results of the present study (with a specific methodological).
Answer2: We appreciate the reviewers' suggestions and modify the discussion.
We modified the first and the second paragraphs accordingly.
We also included additional informations as follows:
Line 250-259: About the heterogeneity of our sample, some differences relating to age and gender must be considered. The reduction in shear strain and increase in thickness of the posterior layer of the TLF in CLBP patients are more significant in the male gender, with a positive correlation between shear strain and low back pain duration [24]. In this sense, range of motion and physical function, body composition, fat distribution pattern, hormonal factors, or structural and/or movement pattern differences between males and females also might be account [24,45]. During aging, extracellular matrix structural, biochemical, cellular and functional changes occur and these age-related alterations in fascial tissues lead to densification and fibrosis (thixotropic behavior of hyaluronic acid and collagen synthesis increasing thickness) contributing to pain occurrence [17,45].

This manuscript is a resubmission of an earlier submission. The following is a list of the peer review reports and author responses from that submission.